# Hub Genes in Non-Small Cell Lung Cancer Regulatory Networks

**DOI:** 10.3390/biom12121782

**Published:** 2022-11-29

**Authors:** Qing Ye, Nancy Lan Guo

**Affiliations:** 1West Virginia University Cancer Institute, West Virginia University, Morgantown, WV 26506, USA; 2Lane Department of Computer Science and Electrical Engineering, West Virginia University, Morgantown, WV 26506, USA; 3Department of Occupational and Environmental Health Sciences, School of Public Health, West Virginia University, Morgantown, WV 26506, USA

**Keywords:** multi-omics networks, hub genes, CRISPR-Cas9, RNAi, proliferation, non-small cell lung cancer, patient survival, biomarkers, therapeutic targets

## Abstract

There are currently no accurate biomarkers for optimal treatment selection in early-stage non-small cell lung cancer (NSCLC). Novel therapeutic targets are needed to improve NSCLC survival outcomes. This study systematically evaluated the association between genome-scale regulatory network centralities and NSCLC tumorigenesis, proliferation, and survival in early-stage NSCLC patients. Boolean implication networks were used to construct multimodal networks using patient DNA copy number variation, mRNA, and protein expression profiles. *T* statistics of differential gene/protein expression in tumors versus non-cancerous adjacent tissues, dependency scores in in vitro CRISPR-Cas9/RNA interference (RNAi) screening of human NSCLC cell lines, and hazard ratios in univariate Cox modeling of the Cancer Genome Atlas (TCGA) NSCLC patients were correlated with graph theory centrality metrics. Hub genes in multi-omics networks involving gene/protein expression were associated with oncogenic, proliferative potentials and poor patient survival outcomes (*p* < 0.05, Pearson’s correlation). Immunotherapy targets *PD1, PDL1, CTLA4*, and *CD27* were ranked as top hub genes within the 10th percentile in most constructed multi-omics networks. *BUB3*, *DNM1L, EIF2S1, KPNB1, NMT1, PGAM1,* and *STRAP* were discovered as important hub genes in NSCLC proliferation with oncogenic potential. These results support the importance of hub genes in NSCLC tumorigenesis, proliferation, and prognosis, with implications in prioritizing therapeutic targets to improve patient survival outcomes.

## 1. Introduction

Non-small cell lung cancer (NSCLC) is the most common cause of cancer mortality for both men and women [1]. It is challenging to manage NSCLC due to its complex somatic mutations and DNA copy number variations (CNV) during cancer genome evolution [2], extensive invasion, acquired therapeutic resistance, and tumor recurrence/metastasis [3]. Recent immunotherapy of blockades of PD1, PDL1, and CTLA4 has improved NSCLC treatment outcomes [4,5] in both neoadjuvant and adjuvant settings for NSCLC of all stages [6,7,8,9,10]. PD1 inhibitor nivolumab [4] is NCCN-recommended for neoadjuvant treatment in combination with chemotherapy for early-stage NSCLC [11]. PDL1 inhibitor atezolizumab is NCCN-recommended for adjuvant immunotherapy following chemotherapy for stage 2/3A NSCLC in patients with PDL1 > 1% [9,10]. Nevertheless, the 5-year survival rate for NSCLC remains a dismal 26% [1]. The molecular mechanisms underlying NSCLC tumorigenesis, proliferation, and recurrence/metastasis are not well-understood. To date, there are no accurate prognostic or predictive biomarkers for optimal treatment selection for individual NSCLC patients. More therapeutic targets are needed to improve NSCLC survival outcomes.

Molecular network analysis is important to understand cancer mechanisms and advance precision oncology [12]. Recent advances in high-throughput technologies empower landscape analysis of molecular machinery at DNA, RNA, and protein levels in tumor initiation, progression, and metastasis. Traditional statistical or machine learning methods merely computing numerical gene associations with clinical outcomes cannot effectively reveal essential molecular interaction networks at multiple regulatory levels. Combined with patient clinical phenotypes, artificial intelligence (AI)-based multi-modal network analysis is needed to embed biological relevance into discovery of biomarkers and therapeutic targets for improved cancer outcomes.

In our previous studies, disease-specific gene co-expression networks were constructed for identification of gene signatures with concurrent crosstalk with major NSCLC signaling hallmarks [13]. These gene signatures led to discovery of a seven-gene panel that can provide patient stratification and prediction of clinical benefits of chemotherapy in early-stage NSCLC patients, including clinical trials [14]. Within the seven-gene panel, *CD27* is an emerging target for cancer immunotherapy [15,16,17,18] involved in *PD1* and *CD70* blockades [19,20,21,22], CD8+ T cell expansion [23], and anti-viral/anti-tumor T cell immunity [24]. As a new generation of immune checkpoint inhibitors (ICIs) [25], CD27 agonist antibodies are being tested as adjuvant therapy in phase I/II clinical trials, showing promising results for multiple tumor types [17,26]. We discovered proliferative multi-omics networks containing *CD27, PD1,* and *PDL1* as well as the seven-gene panel, respectively, implicated in NSCLC prognosis, drug sensitivity, and therapeutics [27,28].

Recent studies showed that hub genes in multi-omics networks are promising cancer biomarkers and therapeutic targets [29,30]. There are insufficient reports on multi-omics network centralities quantified with graph theory metrics and their relevance in cancer etiology and therapy. Genome-scale analysis is needed to evaluate the biological and clinical relevance of network hub genes in NSCLC tumorigenesis, proliferation, and patient survival. In this study, we utilized a computationally efficient Boolean implication algorithm to construct genome-scale multi-omics networks using CNV (*n* = 371), transcriptomics (*n* = 200), and proteomics profiles (*n* = 103) of NSCLC patient bulk tumors. Network centralities were evaluated using graph theory metrics and were correlated with differential gene/protein expression in tumors versus non-cancerous adjacent tissues (NATs), in vitro dependency scores in CRISPR-Cas9/RNAi screening data, and hazard ratios in the Cancer Genome Atlas (TCGA) NSCLC patients (*n* = 1016). Furthermore, the distributions of network centrality metrics of immunotherapy targets *PD1, PDL1, CTLA4,* and *CD27* were evaluated in the constructed NSCLC multi-omics networks.

## 2. Materials and Methods

### 2.1. Boolean Implication Networks

In this study, multi-omics networks were generated with our previously published Boolean Implication network algorithm [31,32]. The details of the application of this algorithm were described in our previous study [28]. Boolean implication networks were used to construct CNV-mediated transcriptional networks in NSCLC tumors as described previously [27,28]. In addition, mRNA co-expression, protein co-expression, and mRNA-mediated protein expression networks were also constructed using the Boolean Implication networks. The implication rules in each network were selected based on the thresholds of their precision and scope [28,31,32]. In this study, the thresholds of precision and scope were calculated using the sample size of each dataset and a *z* value of 1.64 (one-tailed *z* tests, *p* < 0.05, 95% confidence interval).

### 2.2. NSCLC Patient Cohorts

#### 2.2.1. NSCLC Patient Cohort GSE31800

NSCLC patient cohort with the NCBI Gene Expression Omnibus (GEO) accession number GSE31800 [33] contained 271 tumor samples (179 adenocarcinomas and 92 squamous cell carcinomas). All the samples had DNA copy number profiles, among which 49 samples (29 adenocarcinomas, 20 squamous cell carcinomas) had matched microarray gene expression measurements. Gene expression data were generated using the Custom Rosetta-Affymetrix Human platform. Fresh-frozen lung tumors were obtained from Vancouver General Hospital. Microdissection of tumor cells was performed, and total RNA was isolated using RNeasy Mini Kits (Qiagen Inc., Duesseldorf, Germany). Samples were labeled and hybridized to a custom Affymetrix microarray, containing 43,737 probes mapping to approximately 23,000 unique genes, according to the manufacturer’s protocols (Affymetrix Inc., Santa Clara, CA, USA) All data were normalized using the Robust Multichip Average algorithm in R. Of the lung tumor cohort, only samples with sufficient material for RNA isolation were selected for expression analysis.

The genome reference version was converted to hg38. The genome annotation was obtained from the UCSC genome browser with the Python package *cruzdb* on 28 January 2020. Copy number variation (CNV) data were processed with Bioconductor R package “CGHbase” (v1.46.0) [34] and “CGHcall” (v2.48.0) [35]. CNV data were categorized as “1—amplification”, “0—normal”, and “−1—deletion” for constructing CNV co-occurrence (CNV–CNV) networks. The gene expression data were categorized into three categories: “1—up-regulated”, “0—normal”, and “−1—down-regulated” with the method using 27 housekeeping genes described in our previous study [28]. CNV-mediated gene expression networks were built with patients’ gene expression data and their matched CNV profiles. Gene co-expression networks were constructed with the categorized gene expression data.

#### 2.2.2. NSCLC Patient Cohort GSE28582

NSCLC patient cohort with NCBI GEO accession number GSE28582 [36,37] contained 100 tumor samples (50 adenocarcinomas, 22 large cell carcinomas, and 28 squamous cell carcinomas). All samples had SNP array DNA copy number profiles and microarray gene expression data.

A total of 2 μg RNA (RIN value > 7.0) from each tissue specimen was used for analysis on Affymetrix Human Genome U133 Plus 2 arrays (Affymetrix Inc.). Sample preparation, processing, and hybridization were performed according to the GeneChip Expression Analysis Technical Manual (Affymetrix Inc., Rev. 5). Subsequent analyses of the gene expression data were carried out in the freely available statistical computing language R using packages available from the Bioconductor project. The raw data were normalized using the robust multiarray average method and were available in GEO with the accession number GSE28582. Only transcripts with average signal intensities above 5 were used for further analysis. For the comparison of gene expression levels between different patient groups, a two-sided Student’s *t* test was used.

The genome annotation version was converted to hg38. SNP array CNV data were processed with the PennCNV package [38] and were then categorized as “1—amplification”, “0—normal”, and “−1—deletion”. The gene expression data were processed in the same way as described above. CNV–CNV networks, CNV-mediated gene expression networks, and gene co-expression networks were also generated for this patient cohort.

#### 2.2.3. Xu’s Lung Adenocarcinoma (LUAD) Patient Cohort

Xu’s LUAD patient cohort [39] contained paired tumors and non-cancerous adjacent tissues (NATs) samples from 103 Chinese LUAD patients. All samples had protein expression profiles of matched tumors and NATs, among which 51 samples had RNA sequencing gene expression profiles in tumors and 49 matched RNA sequencing gene expression profiles in NATs.

The genome annotation version in Xu’s cohort was hg38 and thus did not require additional conversion. Gene expression and log_10_ transformed protein expression data were categorized into three categories: “1—up-regulated”, “0—normal”, and “−1—down-regulated”. The categorization was based on the distribution of the selected housekeeping genes (*B2M, ESD, FLOT2, GAPDH, GRB2, HPRT1, HSP90AB1, LDHA, NONO, POLR2A, PPP1CA, RHOA, SDCBP*, and *TFRC*) [14,40,41,42]. Gene co-expression networks, mRNA-mediated protein expression networks, and protein co-expression networks of LUAD tumors and NATs were generated for this patient cohort.

#### 2.2.4. TCGA

LinkedOmics (http://www.linkedomics.org/, accessed on 28 April 2021) [43] was utilized to obtain RNA sequencing data of TCGA–LUAD (*n* = 515) and TCGA–LUSC (*n* = 501) patient cohorts. Clinical annotation including survival information was used to calculate the hazard ratios of each gene with a univariate Cox model.

### 2.3. Graph Theory Centrality Metrics

Centrality metrics were used in the network analysis to identify critical nodes. The centrality calculation methods can be divided into two main categories: local and global methods. Local methods detect the influence of nodes based on local information (nodes and their neighbors). These methods require simple information and low computational complexity and are suitable for large and complex networks. Global methods require traversing the global knowledge of the whole network to calculate the impact of nodes. Although the computational complexity is higher, it will obtain some compensation in accuracy and can obtain a more accurate node importance ranking. If the nodes and connected edges in the network change over time, it will be challenging to obtain the global properties. Therefore, global methods are often limited in dynamic situations.

In each of the networks used to calculate the centralities, all implication rule types were merged; i.e., if gene A and gene B have an association with each other in a CNV–CNV network, the amplification of gene A implies the amplification of gene B, and the deletion of gene A can also imply the deletion of gene B. In this study, we only count the two rules as one association for “A implies B”. All the centralities were calculated with the Python package NetworkX [44]. 

#### 2.3.1. Degree Centrality

Degree centrality is a local method that was first proposed for ranking the importance of nodes. Degree centrality is the simplest and most intuitive measure of the importance of a node. In a directed network, degree centrality can be further divided into in-degree centrality and out-degree centrality. In this study, the in- and out-degree centralities for the same-level gene association networks (i.e., CNV–CNV, mRNA co-expression, or protein co-expression) are the same due to the symmetric characteristics. In- and out-degree centralities are summed to degree centrality.

Degree centrality represents the total number of neighbors (the number of edges connected to other nodes) of a node. The more neighbors the more important the node is [45]. A network *G(N, E)* with *N* nodes and *E* edges has an adjacency matrix of *A*. The degree centrality of node *i* in *G(N, E)* can be expressed as: (1)$$C_{D}\left(i\right)=\frac{\sum_{j=1}^{N}A_{ij}}{N-1}$$

*j* (*j* ≠ *i*) denotes all other nodes in the network, and *A_ij_* is the value in adjacency matrix A. If there is a connection between node *i* and node *j*, then *A_ij_* = 1; otherwise, *A_ij_* = 0. $\overset{N}{\underset{j=1}{\sum }}A_{ij}$ represents the total number of neighbors (connections) of node *i*. *N-1* is the maximum number of possible connections of a single node in the network.

#### 2.3.2. Eigenvector Centrality

Eigenvector centrality takes into account not only the number of neighbors of a node but also the importance of its neighbors in the network [46]. The main idea of eigenvector centrality is that each node in the network is assigned a centrality, and the centrality of each node is the sum of the centrality of its neighbors to which it is connected. A node will have its centrality boosted by connecting to high-centrality nodes [47]. Nodes with higher centrality can be connected to a large number of general nodes or a small number of other nodes with high centrality.

A network *G(N, E)* with *N* nodes and *E* edges has adjacency matrix *A*. Similar to degree centrality, *A_ij_* is the value in adjacency matrix *A*. If there is a connection between node *i* and node *j*, then *A_ij_* = 1, and vice versa *A_ij_* = 0. Eigenvector centrality of node *i* is expressed as [48]:(2)$$C_{E}\left(i\right)=\frac{1}{\lambda }\underset{j\in M\left(i\right)}{\sum }A_{ij}x_{j}$$
where *λ* is a constant representing the maximum value of the eigenvalues of the adjacency matrix *A*. *M(i)* is the set of neighboring nodes of node *i*. *x_i_* is the score of the importance of node *i*, *x* = [*x*_1_*, x*_2_*, x*_3_*, …, x_n_*]*^T^*, then Equation (2) can be written as the eigenvector equation *Ax* = *λx*.

The basic way to calculate the vector x is to give an initial *x*(0) value, usually, 1, multiply the vector *x* cyclically with *A*, and update *x* with the following Equation (3) until *x* stabilizes and does not change, then the final value of *x* is obtained. If *x* is divided by the principal eigenvalue λ of adjacency matrix *A* during each iteration, the equation yields a convergent non-zero solution, i.e., *x* =*λ*^−1^*Ax*
(3)$$x\left(t\right)=\left(\lambda -1\right)Ax\left(t-1\right),\;t=1,\;2,\;3,\;\ldots$$

#### 2.3.3. Betweenness Centrality

Betweenness centrality considers that the more times a node is present in the shortest path between any two non-adjacent nodes, the node is routable and more important in the network. Betweenness centrality is a global method of computing centrality, which requires first getting all the shortest paths in the network. If a node appears on the shortest path of all node pairs in the network more often, then that node is more important. The network *G(N, E)* with *N* nodes and *E* edges, and the set *V* denotes the set of all nodes in the network. The betweenness centrality of node *i* in *G(N, E)* can be expressed as [49]:(4)$$C_{B}\left(i\right)=\frac{2}{\left(N-1\right)\left(N-2\right)}\underset{s\neq i\neq t\in V}{\sum }\frac{p_{st}\left(i\right)}{p_{st}}$$

*s* and *t* are any two nodes in the network, the two nodes cannot be the same or node *i*. $p_{st}$ denotes the number of shortest paths between node *s* and node *t* in the network, and $p_{st}\left(i\right)$ denotes the number of entries in the shortest path between node *s* and node *t* that passed through node *i*. The term $\frac{2}{\left(N-1\right)\left(N-2\right)}$ is used for normalization.

#### 2.3.4. Closeness Centrality

Closeness centrality is also a global method based on the shortest path between nodes. It ranks nodes based on the average distance between the target node and other nodes in the network. The smaller the average distance between a node and other nodes, the greater the closeness centrality of that node, i.e., the more critical that node is. *d_ij_* denotes the length of the shortest path between any two nodes in the network, then $d_{i}=\frac{1}{N-1}\underset{j\neq i}{\sum }d_{ij}$ is the average shortest path length from node *i* to other nodes in the network. The closeness centrality of node *i* is expressed as follows:(5)$$C_{C}\left(i\right)=\frac{1}{d_{i}}=\frac{N-1}{\sum_{j\neq i}d_{ij}}$$

This formula is only applicable to the case of a connected network (i.e., there exists a path from every node to every other node in the network).

#### 2.3.5. VoteRank Centrality

VoteRank centrality is an algorithm proposed by Zhang et al. [50] to identify important nodes based on the phenomenon of voting in reality. The VoteRank algorithm simulates the voting process by considering that each node has two attributes: voting score (*VS*) and voting ability (*VA*). The sum of *VA*s of all neighbors of node *i* is the *VS* of node *i*. That is, in a network *G(N, E)* with *N* nodes and *E* edges, the VoteRank centrality of node *i* is:(6)$$C_{V}\left(i\right)=\underset{j\in M\left(i\right)}{\sum }VA_{j}$$

*M*(*i*) is the set of neighboring nodes of node *i*. The VoteRank algorithm selects one node that gets the highest voting score in each round. If the first *r* nodes need to be selected, *r* rounds of operation need to be performed, and the order of the nodes in the result set *S* is their VoteRank centrality. 

### 2.4. CRISPR-Cas9 Knockout Assays

In the Cancer Cell Line Encyclopedia (CCLE) panel, the dependency scores of whole-genome CRISPR-Cas9 knockout screening data of 94 human NSCLC cell lines were obtained from the DepMap portal (https://depmap.org/portal/download/all/, accessed on 12 September 2022; release 21Q4) [51,52]. The dependency score threshold used for determining a significant effect on a cell line was –0.5 in this study [27,53]. A gene with a dependency score lower than –0.5 was considered as having a significant effect of CRISPR-Cas9 knockout on the corresponding cell line.

### 2.5. RNAi Knockdown Assays

The dependency scores of whole-genome RNA interference (RNAi) knockdown screening data of 92 human NSCLC cell lines in CCLE were also obtained from the DepMap portal (https://depmap.org/portal/download/all/, accessed on 12 September 2022; release 21Q4) [51,52]. The dependency score threshold used for determining a significant effect on the cell line was also −0.5 [27,53]. A dependency score smaller than −0.5 indicated the gene has a significant RNAi knockdown effect on the corresponding cell line. 

### 2.6. Statistical Methods

Statistical analysis was performed in R software (version 4.1.3) with RStudio (version 2022.07.2 Build 576). The comparisons of two groups, such as differential expression analysis, were performed with two sample *t* tests. Univariate Cox proportional hazards regression was performed to obtain hazard ratios using the R package “*survival*”. To test if a constructed Boolean implication network had higher average centrality values compared with random networks, the averaged centrality metrics (except VoteRank centrality) of the constructed network were compared with those of 1000 randomly selected networks with the same number of genes. The random networks contained the same number of genes randomly selected from the whole genome excluding our identified network genes. The *p* values were determined as the percentage of the constructed network that did not have a higher average centrality value than a random network. Comparison of VoteRank centrality was performed with one-tailed two-sample Wilcoxon tests. A total of 1000 Wilcoxon tests were applied in the random tests, and the *p* values showed a percentage of non-significant results. The correlation between centrality metrics and tumorigenesis, proliferation, and cancer patient survival outcomes was measured with Pearson’s correlation coefficients in the genome-scale. *T* statistics of two-tailed two-sample *t* tests (unpaired) in differential expression between tumor versus NATs were used for tumorigenesis assessment. Positive *t* statistics indicated higher expression in tumors than in NATs and vice versa. Gene dependency scores of human NSCLC cell lines from CRISPR-Cas9/RNAi data were used in proliferation analysis. Negative dependency scores indicated the cancer cell line growth was highly dependent on the gene; positive dependency scores indicated the cell line grew faster after the gene was knocked out/knocked down. Univariate hazard ratios were used in the association assessment with patient survival outcomes. Hazard ratios higher than 1 indicate increased risks from tumor recurrence, metastasis, or death from disease. Since the VoteRank centrality indicated a higher rank with a smaller number, which was the reverse of the other centrality metrics, a positive correlation with VoteRank was equivalent to a negative correlation with other centrality metrics. Any statistical results with a *p* value < 0.05 were considered significant. 

ToppFun, an online tool from ToppGene Suite [54], was used to perform the functional enrichment analysis. The ToppFun tool can be accessed at https://toppgene.cchmc.org/enrichment.jsp (accessed on 25 November 2022).

## 3. Results

### 3.1. Multi-Omics Networks of NSCLC Patient Cohorts

Using the Boolean Implication network algorithm, 12 multi-omics networks were constructed, including CNV–CNV networks, CNV-mediated gene expression (GE) networks, and mRNA co-expression networks for patient cohorts GSE28582 (*n* = 100) [36,37] and GSE31800 (*n* = 271) [33], respectively; mRNA co-expression networks, mRNA-mediated protein expression networks, and protein co-expression networks in tumors and NATs samples, respectively, in Xu’s LUAD patient cohort [39]. Detailed network information was provided in Table 1. 

### 3.2. Association of Centrality Metrics with Tumorigenesis, Proliferation, and Patient Survival

The centrality metrics in this study were generated based on the 12 multi-omics networks shown in Table 1. Degree centrality, in-degree centrality, out-degree centrality, eigenvector centrality, betweenness centrality, closeness centrality, and VoteRank centrality were calculated for each network. To assess the association between multi-omics network centrality and NSCLC tumorigenesis, proliferation, and patient survival in the genome scale, correlation coefficients between the seven centrality metrics and *t* statistics of differential gene/protein expression in tumors versus NATs, dependency scores in CRISPR-Cas9/RNAi, and hazard ratios in univariate Cox model of survival analysis were computed for each of the 12 multi-omics networks. Figure 1 showed the number of concordant significant correlations with seven centrality metrics between each pair of networks. Selected hub genes were provided in Appendix A. Measurements used to assess tumorigenesis, proliferation, and patient survival were provided in Appendix A. Categorized data as input to generate multi-omics networks were provided in Appendix A. 

Differential mRNA expression in tumors versus NATs in Xu’s LUAD patients [39] had a concordant significant correlation with network centrality metrics across independent patient cohorts except for CNV–CNV networks (Table 2). In CNV-mediated GE networks and mRNA co-expression networks constructed in GSE28582 and GSE31800 patient cohorts, genes with higher network centrality, quantified with multiple metrics, correlated with higher mRNA expression in tumors. For mRNA co-expression networks, mRNA-mediated protein expression networks, and protein co-expression networks constructed in both tumors and NATs, higher network centrality metrics correlated with higher mRNA expression in tumors. These results indicate that hub genes in multi-omics networks in tumors and NATs appear to be oncogenic. 

Differential protein expression in tumors versus NATs in Xu’s LUAD patients [39] had a concordant significant correlation with centrality metrics in protein co-expression networks in both tumors and NATs, respectively (Table 3), consistent with elevated mRNA expression of hub genes in tumors in the above multi-omics networks (Table 2). These results suggest that hub genes in protein co-expression networks in both tumors and NATs have higher oncogenic potential. Interestingly, in CNV–CNV networks constructed in both GSE28582 and GSE31800, hub genes with more co-occurrence of CNV in NSCLC tumors were associated with lower protein expression in tumors, suggesting tumor-suppressive potential (Table 3).

Next, we assessed the association between multi-omics network centrality and NSCLC proliferation. In genome-scale CRISPR-Cas9/RNAi screening, hub genes in CNV–CNV networks were associated with higher dependency scores, i.e., anti-proliferative potential (Table 4 and Table 5), consistent with a lower protein expression in tumors and putative tumor-suppressive potential (Table 3). In contrast, hub genes in CNV-mediated GE networks, mRNA co-expression networks, and mRNA-mediated protein expression networks in both tumors and NATs correlated with lower dependency scores, i.e., proliferation, across different NSCLC patient cohorts in CRISPR-Cas9/RNAi screening (Table 4 and Table 5). It is noteworthy that regulated genes represented with higher in-degree centrality were associated with proliferative potential, whereas regulatory genes represented with higher out-degree centrality were associated with anti-proliferative potential in CNV-mediated GE networks and mRNA-mediated protein expression networks. Hub genes in protein co-expression networks in NATs, measured with multiple metrics, appeared to be more proliferative in human NSCLC cell lines (Table 4 and Table 5), consistent with their putative oncogenic potential observed in Table 2 and Table 3. 

The association between network centralities and NSCLC patient survival was also examined. Hazard ratios in univariate Cox modeling of combined TCGA–LUAD (*n* = 515) and TCGA–LUSC (*n* = 501) were used in the genome-wide evaluation. Hub genes in mRNA co-expression networks, mRNA-mediated protein expression, and protein co-expression networks were associated with higher hazard ratios in multiple patient cohorts, suggesting they are survival hazard genes (Table 6). These results are consistent with the oncogenic and proliferative potential of hub genes described above. Regulatory genes with higher out-degree centralities in mRNA-mediated protein expression networks in tumors and NATs in Xu’s LUAD cohort [39] were associated with lower hazard ratios in TCGA patients. These results are consistent with the anti-proliferative potential of regulatory genes (Table 4 and Table 5). The association of VoteRank in mRNA co-expression networks tumors and NATs in Xu’s Chinese patient cohort was inconsistent with those of other centrality metrics in GSE31800 and GSE28582. Overall, hub genes in multi-omics networks tend to be associated with increased survival hazards, i.e., poor prognosis, in NSCLC patients (Table 6). 

### 3.3. Distributions of Multi-Omics Network Centrality Metrics of Therapeutic Targets

Having substantiated the association between multi-omics network centralities and NSCLC tumorigenesis, proliferation, and patient survival, we sought to investigate the potential of hub genes as therapeutic targets. Here, we examined four established immune checkpoint inhibitors (ICIs) for NSCLC immunotherapy, including *PD1, PDL1, CD27*, and *CTLA4*. The percentile of these ICIs was determined for the seven centrality metrics of twelve constructed multi-omics networks. Figure 2 showed the rank of centrality metrics of *CD27, CTLA4, PD1*, and *PDL1* that were within the top 10th percentile in our constructed multi-omics networks. These ICIs were top hub genes in CNV-mediated gene expression networks in GSE28582 and GSE31800, mRNA co-expression networks in tumors from GSE28582, GSE31800, and Xu’s LUAD patient cohort [39], and mRNA-mediated protein expression network of tumors in Xu’s LUAD patient cohort [39]. These ICIs were not ranked within the top 10th percentile of the examined centrality metrics in CNV–CNV networks or protein co-expression networks constructed in this study. These results imply that established therapeutic targets in immunotherapy are often top-ranked hub genes in multi-omics networks in tumors across NSCLC patient cohorts. 

### 3.4. Clinical Relevance of Multi-Omics Network Centrality 

We utilized Boolean implication networks and identified two multi-omics networks implicated in NSCLC proliferation, prognosis, and drug sensitivity in our previous studies [27,28]. Both multi-omics networks led to discovery of novel therapeutic targets for treating NSCLC [27,28]. Here, we examined if these two clinically relevant multi-omics networks had higher network centralities in the genome-scale compared with 1000 random networks with the same number of genes. Both networks utilized CNV and GE profiles. The genes included in these two networks were provided in Appendix A.

The results showed that, in genome-scale CNV–CNV networks in GSE28582 and GSE31800, the genes from network A and network B did not have significantly higher average centrality measurements than randomly selected gene sets. In CNV-mediated GE networks in GSE28582 and GSE31800, the genes from network A and network B both had significantly (*p* < 0.05) higher average in-degree centrality, closeness centrality, and betweenness centrality values than randomly selected sets of genes. In gene co-expression networks in GSE28582 and GSE31800, the genes from network A and network B had significantly (*p* < 0.05) higher averaged centralities than randomly selected sets of genes in almost all the evaluated metrics (Figure 3). These results show that multi-omics networks with clinical relevance tend to contain more hub genes than randomly selected gene sets from genome-scale CNV-mediated GE networks and gene co-expression networks. 

### 3.5. Important Hub Genes in NSCLC 

To select hub genes important in NSCLC, we first extracted the genes that ranked within the top 10th percentile for all seven evaluated centrality metrics in at least one of the 12 multi-omics networks. Then, the measurements of tumorigenesis (*t* statistics of mRNA and protein differential expression in tumors vs. NATs in Xu’s LUAD cohort [39]) and patient survival (hazard ratios in univariate modeling of mRNA expression in TCGA NSCLC patients) were extracted for each gene. The genes that were significant (*p* < 0.05) in at least one measurement were shown in Appendix A.

Table 7 showed the hub genes that were significant and concordant in all measurements of tumorigenesis and patient survival. These genes had significantly higher mRNA and protein expression in tumors compared with NATs and had an increased hazard ratio (>1) in patient survival. These genes are potential oncogenes in NSCLC. Among these genes, *BUB3*, *DNM1L, EIF2S1, KPNB1, NMT1, PGAM1,* and *STRAP* had a significant dependent score in at least 41 human NSCLC cell lines tested in CRISPR-Cas9 or RNAi screening, indicating they are also proliferation genes. Venn diagrams of gene associations involving these seven genes in NSCLC regulatory networks were provided in Appendix A. Among these NSCLC regulatory networks, *CDC6* and *DIAPH3* had significant mRNA co-expression (*p* < 0.05, *z* tests) with all seven genes in both GSE28582 [36,37] and Xu’s LUAD tumors [39] (Figure 4). Twenty proteins had significant co-expression (*p* < 0.05, *z* tests) with the protein expression of all seven genes (Figure 4). Significantly enriched cytobands and gene families of this network were obtained with ToppFun and were listed in Appendix A.

## 4. Discussion

NSCLC is the leading cause of cancer-related deaths due to its complex etiology. Use of small-molecule tyrosine kinase inhibitors (TKIs) and immunotherapy has clinically benefited selected NSCLC patients [55]. Nevertheless, the overall cure and survival rates of NSCLC remain low. Novel biomarkers and drug targets are needed to improve patient care outcomes. The availability of multimodal data offers emerging opportunities for the discovery of biomarkers and therapeutic targets for better cancer outcomes in broader patient populations. 

The molecular machinery in a tumor and its microenvironment involves complicated interactions among genes and proteins functioning in epithelial, immune, and stromal cells as well as other systemic host factors [56]. Given this intricacy, multi-omics networks that integrate these elements should be elucidated to better understand tumor biology and molecular mechanisms for development of novel therapeutic strategies. Recent multi-omics studies identified several hub genes as cancer biomarkers and drug targets, including NSCLC [29,30]. Nevertheless, it is not unknown if molecular network centralities are associated with tumorigenesis, proliferation, and patient survival in NSCLC in an unbiased, systematic evaluation. 

A barrier to evaluating genome-scale network centralities lies in that current computational methods have certain limitations in modeling multi-omics networks. Correlation networks (relevance networks) [57] cannot integrate continuous expression variables with discrete data, such as CNV. Bayesian networks are topologically acyclic and cannot model cyclic molecular interactions [58]. More importantly, probabilistic graphical models, including Bayesian networks and Markov networks [59], describe joint probability distribution and have exponential complexity [60], making it impossible to model genome-scale networks. Other Boolean networks [61] use Fisher’s exact tests or *χ*^2^-square tests to analyze binary variables in quadrants that do not present multivariate biological states.

This study utilized our developed Boolean implication networks to construct genome-scale multi-omics networks. Our Boolean implication network algorithm is based on prediction logic and overcomes the theoretical limitations of these models, with its capability to efficiently analyze multivariate biological data, cyclic molecular interactions, and discrete and continuous multi-omics data in seamless integration [27,28]. Our Boolean implication networks revealed more biologically relevant molecular interactions in NSCLC tumors than other Boolean networks, Bayesian networks, and correlation networks in comprehensive evaluation using MSigDB [32]. Using our Boolean implication networks, novel gene signatures co-expressed with major NSCLC signaling hallmarks were identified as prognostic of NSCLC survival outcomes, which outperformed the existing gene signatures in the same patient data [13]. Furthermore, a prognostic and predictive seven-gene panel was discovered from these identified candidate genes and was confirmed in qRT-PCR [14], RNA-sequencing data of TCGA [27], and proteomic profiles in more than 1600 NSCLC patients, including a clinical trial JBR.10.

This study conducted a landscape evaluation of the biological and clinical relevance of multi-omics Boolean implication network centralities rigorously quantified with graph theory metrics in NSCLC tumors. Our results across multiple patient cohorts showed that hub genes in CNV-mediated GE networks, mRNA co-expression networks, mRNA-mediated protein expression networks, and protein co-expression networks in NSCLC tumors had oncogenic and proliferative potential and were associated with poor patient prognosis. Hub genes in protein co-expression networks in NATs also seemed to be more proliferative and oncogenic in NSCLC. Regulated genes represented with higher in-degree centrality in multi-omics networks were associated with proliferative potential and worse patient survival, whereas regulatory genes represented with higher out-degree centrality were associated with anti-proliferative potential and better patient survival. The results on CNV co-occurrence networks were different from those on multi-omics networks involving gene/protein expression; hub genes with more co-occurrences of CNV in NSCLC tumors appeared to have tumor-suppressive and anti-proliferative potential. 

Our previous studies identified two CNV-mediated GE networks containing proliferative and prognostic gene signatures, capable of providing accurate patient stratification in more than 1000 NSCLC patients [27,28]. One multi-omics network contains 66 genes, including the prognostic and predictive seven-gene panel [27], and the other one with 30 genes involves *PD1, PDL1*, and *CD27* [28]. In the TCGA consortium, the seven marker genes and major ICIs have more CNV aberrations than mutations in NSCLC tumors. In addition to their prognostic capacity, these two multi-omics networks can determine drug sensitivity to 10 therapeutic regimens in 135 human NSCLC cell lines [27,28]. Further analysis of these two multi-omics networks led to the discovery of novel targeted therapies as new or repositioning drugs for treating NSCLC [27,28]. Both networks had significantly higher average centrality than random networks selected in genome-wide CNV-mediated GE networks and gene co-expression works. These results substantiate that clinically relevant multi-omics networks have more hub genes than random networks. In addition, NSCLC immunotherapy targets, including *PD1, PDL1, CTLA4*, and *CD27*, also ranked as top hub genes in most multi-omics networks constructed in this study. To show the relevance of network centrality and therapeutic targets, Figure 2 included immunotherapy targets that are either used for treating NSCLC patients (including PD1, PDL1, and CTLA4) or showed promising results in phase I/II clinical trials (CD27). Many NSCLC biomarkers, including *CD151,* that have not been substantiated in clinical trials were not included in Figure 2. *CD151,* a cancer driver and tumor metastasis promoter [62,63,64,65,66], was ranked as a top hub gene within the 10th percentile of degree centrality in the CNV-mediated gene expression network in GSE28582 [36,37] and mRNA-mediated protein expression network in Xu’s LUAD NATs [39]. These results further support the importance of hub genes in NSCLC therapeutics.

This study also identified important hub genes in NSCLC tumor cell proliferation and oncogenic processes. *BUB3* is within the spindle assembly checkpoint (SAC) complex. The BUB3 protein is essential in activation of the SAC complex, which, in turn, regulates meiosis and causes mitotic arrest [67]. *BUB3* up-regulation was found in multiple human cancers, including NSCLC, and was linked to poor prognoses [67]. The *DNM1L* gene encodes dynamin-related protein 1 (DRP1), which regulates mitochondria fission [68]. DRP1, highly expressed in *Kras*-mutant NSCLC, is critical in tumor cell proliferation through utilization of lactate in the metabolic reprogramming of NSCLC [69]. Inhibition of DRP1 and NRF2 restored cisplatin sensitivity and stopped the spread of cancer cells in a mouse model of metastatic breast cancer cells latent in the lung soft tissue [70]. Stabilization of oncoprotein EIF2S1 diminished the efficacy of EGFR TKIs in NSCLC treatment through binding of lncRNA LCETRL4 [71]. *KPNB1* promoted NSCLC proliferation by mediating nuclear translocation of PDL1 via the *Gas6/MerTK* signaling pathway [72]. Down-regulation of *KPNB1* induced by *PLK1* inhibition caused apoptosis in lung adenocarcinoma [73]. *NMT1* was overexpressed in spheroid cells, NSCLC tumors, and patients with poor survival outcomes [74]. *NMT1* promoted stemness in NSCLC via activating the PI3K/AKT pathway. *NMT1* also accelerated NSCLC tumor metastasis and resistance to cisplatin [74]. Oncogenic *STRAP* [75] inhibits E-cadherin and *P21*(*CIP1*) through modulation of transcription factor *SP1,* contributing to tumor progression [76]. *GALNT2* functions as an oncogenic driver in NSCLC proliferation, migration, and invasion in vitro, and its knockdown restrained tumor formation in vivo [77]. *PFKP,* involved in metabolism, is a suggested oncogene in lung cancer [78]. *PTGES3* correlates with poor patient prognosis and immune infiltrates in lung adenocarcinoma [79] and is an oncogenic driver within a 10-gene metabolic panel in NSCLC [80]. Overall, the literature supports that the 10 hub genes (Table 7) identified in this study are potential oncogenes in NSCLC. This study shows that multi-omics network centrality can be used as a prioritization method in selection of biomarkers and therapeutic targets. Hub genes can be candidate genes for development of clinical diagnostic tests. The final determination of inclusion of the candidate genes in clinical tests will be made based on the assay optimization and validation results in multiple patient cohorts according to REMARK guidelines [81,82]. 

## 5. Conclusions

To the best of our knowledge, this study is the first systematic revelation of the association between multi-omics network centralities and NSCLC tumorigenesis, proliferation, and patient survival. Hub genes in multimodal networks involving gene/protein expression tended to be more oncogenic, proliferative, and hazardous for patient survival. Hub genes with more co-occurrences of CNV aberrations appeared to be tumor-suppressive and anti-proliferative. Regulated genes in hubs were associated with proliferative potential and worse patient survival, whereas regulatory genes in hubs were associated with anti-proliferative potential and better patient survival. Immunotherapy targets, including *PD1, PDL1, CTLA4*, and *CD27*, were top hub genes in the majority of the constructed multi-omics networks in NSCLC tumors. *BUB3*, *DNM1L, EIF2S1, KPNB1, NMT1, PGAM1,* and *STRAP* were discovered as important hub genes in NSCLC proliferation with oncogenic potential. These results contributed to a better understanding of NSCLC tumor biology and underlying mechanisms. This study showed that gene centrality metrics in multi-omics networks can be used in prioritization of candidates for biomarkers and drug targets. The AI/big data technology presented in this study can be applied to many other human cancers.

## 6. Patents

Our AI technology using Boolean implication networks for discovery of biomarkers and therapeutic targets is included in patent PCT/US22/75136.

## Figures and Tables

**Figure 1 biomolecules-12-01782-f001:**
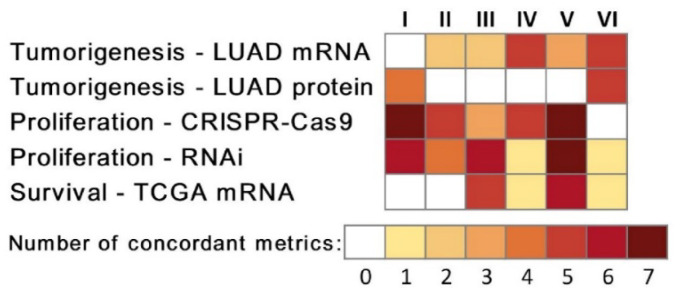
Concordance of correlation coefficients between seven centrality metrics of the selected networks with NSCLC tumorigenesis, proliferation, and patient survival. Tumorigenesis was described with the *t* statistics (two-sample *t* tests) of tumor vs. NAT differential expression in mRNA (*n*_tumor_ = 51, *n*_NAT_ = 49) and protein (*n*_tumor_ = *n*_NAT_ = 103) datasets in Xu’s LUAD patients [39]. Proliferation was assessed in human NSCLC cell lines with dependency scores in in vitro CRISPR-Cas9 (*n* = 94) and RNAi (*n* = 92) genome-wide screening. Patient survival was represented by hazard ratios in univariate Cox modeling of TCGA RNA sequencing data of NSCLC patient tumors (*n* = 1016). Each cell in the figure showed the number of metrics with concordant significant Pearson’s correlation coefficients in a pair of compared networks: I. CNV–CNV networks (GSE28582 and GSE31800); II. CNV-mediated GE networks (GSE28582 and GSE31800); III. gene co-expression networks (GSE28582 and GSE31800); IV. gene co-expression networks (Xu’s LUAD tumors and NATs [39]); V. mRNA-mediated protein expression networks (Xu’s LUAD tumors and NATs [39]); VI. protein co-expression networks (Xu’s LUAD tumors and NATs [39]).

**Figure 2 biomolecules-12-01782-f002:**
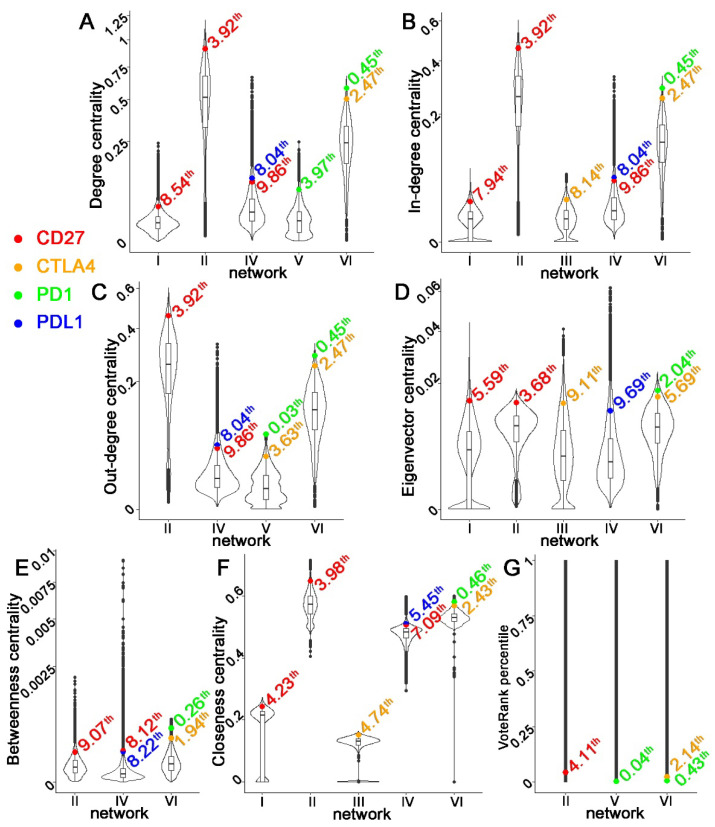
Distributions of centrality metrics in multi-omics networks with *CD27, CTLA4, PD1*, or *PDL1* ranked within the top 10th percentile. Each subplot represented a centrality metric. (**A**) Degree centrality. (**B**) In-degree centrality. (**C**) Out-degree centrality. (**D**) Eigenvector centrality. (**E**) Betweenness centrality. (**F**) Closeness centrality. (**G**) VoteRank centrality. Each violin plot showed the distribution of the centrality metric in one specific network: I. CNV-mediated gene expression network in GSE28582; II. mRNA co-expression network in GSE28582; III. CNV-mediated gene expression network in GSE31800; IV. mRNA co-expression network in GSE31800; V. mRNA-mediated protein expression network in tumors of Xu’s LUAD patient cohort [39]; VI. mRNA co-expression network in tumors of Xu’s LUAD patient cohort [39].

**Figure 3 biomolecules-12-01782-f003:**
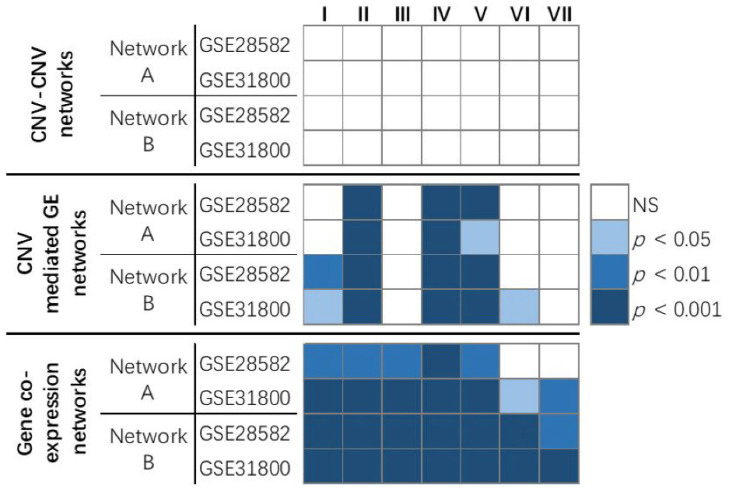
The comparison of centrality metrics of two published multi-omics networks vs. randomly selected networks with the same number of genes. Network A contains 30 genes in the *CD27, PD1*, and *PDL1* multi-omics network in NSCLC tumors [28]. Network B contains 66 genes in the multi-omics network of the 7-gene prognostic signature in NSCLC tumors [27]. The *p* values showed the percentage of randomly selected genes having a larger averaged centrality (except VoteRank) than networks A and B. The *p* value of VoteRank centrality showed the percentage of randomly selected genes having a lower averaged rank (one-tailed Wilcoxon rank sum test, *p* < 0.05) than networks A and B. Each column showed a centrality metric: I. degree centrality; II. in-degree centrality; III. out-degree centrality; IV. eigenvector centrality; V. closeness centrality; VI. betweenness centrality; VII. VoteRank centrality.

**Figure 4 biomolecules-12-01782-f004:**
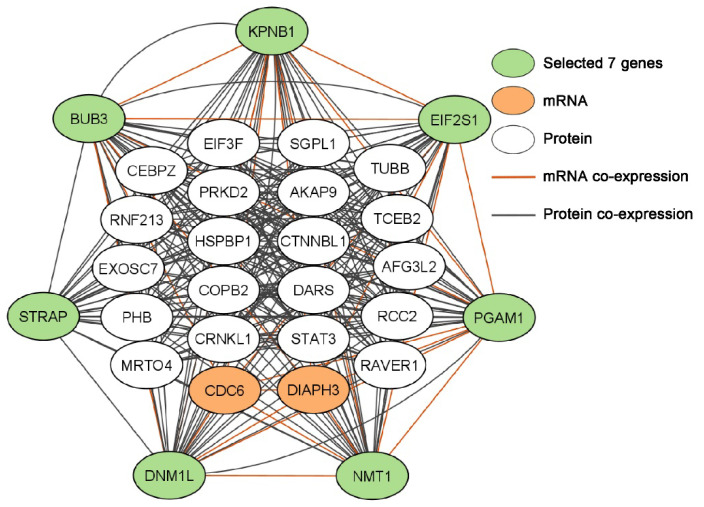
Gene and protein co-expression network of selected seven genes (*BUB3*, *DNM1L, EIF2S1, KPNB1, NMT1, PGAM1,* and *STRAP*) in NSCLC tumors.

**Table 1 biomolecules-12-01782-t001:** Information of multi-omics networks in tumor samples from non-small cell lung cancer patients. The network nodes are genes and network edges are computed gene associations (one-tailed *z* tests, *p* < 0.05, 95% confidence interval).

Patient Cohort	Network (Number of Patient Samples)	Number of Network Nodes	Number of Network Edges
GSE28582 [37,38]	CNV–CNV (*n* = 100)	11,533	3,228,054
	CNV-mediated GE (*n* = 100)	20,836	3,102,789
	mRNA co-expression (*n* = 100)	15,297	48,373,448
GSE31800 [34]	CNV–CNV (*n* = 271)	19,344	20,950,447
	CNV-mediated GE (*n* = 49)	17,442	2,421,110
	mRNA co-expression (*n* = 49)	15,180	4,541,858
Xu’s LUAD [40]	NATs: mRNA co-expression (*n* = 49)	12,408	20,419,308
	NATs: mRNA-mediated protein expression (*n* = 49)	13,254	436,488
	NATs: Protein co-expression (*n* = 103)	2206	785,204
	Tumors: mRNA co-expression (*n* = 51)	11,938	16,101,406
	Tumors: mRNA-mediated protein expression (*n* = 51)	13,047	1,501,406
	Tumors: Protein co-expression (*n* = 103)	3072	2,273,792

**Table 2 biomolecules-12-01782-t002:** Correlations between seven centralities of the selected networks and *t* statistics of differential mRNA expression in tumors vs. NATs in Xu’s LUAD cohort [39]. pos: Pearson’s correlation coefficient *r* > 0 and *p* < 0.05; neg: *r* < 0 and *p* < 0.05. The blue color indicates concordant positive correlations.

Tumorigenesis—Differential mRNA Expression in Tumors vs. NATs (*n* = 51)		Degree Centrality	In-Degree Centrality	Out-Degree Centrality	Eigenvector Centrality	Betweenness Centrality	Closeness Centrality	VoteRank Centrality
CNV–CNV networks	CNV–CNV network (GSE28582, *n =* 100)	neg	neg	neg	-	neg	-	pos
	CNV–CNV network (GSE31800, *n =* 271)	-	-	-	-	-	-	-
CNV-mediated GE networks	CNV-mediated GE network (GSE28582, *n =* 100)	-	pos	neg	pos	-	pos	pos
	CNV-mediated GE network (GSE31800, *n =* 49)	pos	pos	pos	pos	pos	-	neg
mRNA co-expression networks	mRNA co-expression network (GSE28582, *n =* 100)	pos	pos	pos	pos	pos	pos	-
	mRNA co-expression network (GSE31800, *n =* 49)	-	-	-	pos	-	pos	-
mRNA co-expression networks in Xu’s LUAD tumors and NATs	mRNA co-expression network in LUAD tumors (*n*=51)	pos	pos	pos	pos	-	pos	-
	mRNA co-expression network in LUAD NATs (*n*=49)	pos	pos	pos	pos	pos	pos	neg
mRNA-mediated protein expression networks in Xu’s LUAD tumors and NATs	mRNA-mediated protein expression network in LUAD tumors (*n*=51)	pos	pos	-	pos	pos	pos	pos
	mRNA-mediated protein expression network in LUAD NATs (*n =* 49)	pos	pos	-	-	-	pos	-
Protein co-expression networks in Xu’s LUAD tumors and NATs	Protein co-expression network in LUAD tumors (*n =* 103)	pos	pos	pos	pos	pos	pos	-
	Protein co-expression network in LUAD NATs (*n =* 103)	pos	pos	pos	pos	-	pos	-

**Table 3 biomolecules-12-01782-t003:** Correlations between seven centralities of the selected networks and *t* statistics of differential protein expression in tumors vs. NATs in Xu’s LUAD cohort (*n* = 103) [39]. pos: Pearson’s correlation coefficient *r* > 0 and *p* < 0.05; neg: *r* < 0 and *p* < 0.05. The blue color indicates concordant positive correlations. The orange color indicates concordant negative correlations.

Tumorigenesis—Differential Protein Expression in Tumors vs. NATs (*n* = 103)		Degree Centrality	In-Degree Centrality	Out-Degree Centrality	Eigenvector Centrality	Betweenness Centrality	Closeness Centrality	VoteRank Centrality
CNV–CNV networks	CNV–CNV network (GSE28582, *n =* 100)	neg	neg	neg	-	neg	neg	pos
	CNV–CNV network (GSE31800, *n =* 271)	neg	neg	neg	neg	neg	-	-
CNV-mediated GE networks	CNV-mediated GE network (GSE28582, *n =* 100)	-	pos	neg	pos	-	pos	pos
	CNV-mediated GE network (GSE31800, *n =* 49)	-	-	-	-	-	-	-
mRNA co-expression networks	mRNA co-expression network (GSE28582, *n =* 100)	pos	pos	pos	pos	-	pos	-
	mRNA co-expression network (GSE31800, *n =* 49)	neg	neg	neg	neg	neg	-	pos
mRNA co-expression networks in Xu’s LUAD tumors and NATs	mRNA co-expression network in LUAD tumors (*n =* 51)	-	-	-	-	neg	-	pos
	mRNA co-expression network in LUAD NATs (*n =* 49)	pos	pos	pos	pos	pos	pos	neg
mRNA-mediated protein expression networks in Xu’s LUAD tumors and NATs	mRNA-mediated protein expression network in LUAD tumors (*n =* 51)	pos	pos	neg	pos	pos	pos	pos
	mRNA-mediated protein expression network in LUAD NATs (*n =* 49)	-	-	-	-	-	-	-
Protein co-expression networks in Xu’s LUAD tumors and NATs	Protein co-expression network in LUAD tumors (*n =* 103)	pos	pos	pos	pos	-	pos	-
	Protein co-expression network in LUAD NATs (*n =* 103)	pos	pos	pos	pos	-	pos	-

**Table 4 biomolecules-12-01782-t004:** Correlations of seven centrality metrics of selected networks with CRISPR-Cas9 dependency scores. pos: Pearson’s correlation coefficient *r* > 0 and *p* < 0.05; neg: *r* < 0 and *p* < 0.05; -: not significant. The numbers in parentheses showed the number of NSCLC cell lines with a significant correlation coefficient. The blue color indicates concordant positive correlations. The orange color indicates concordant negative correlations.

Proliferation-CRISPR-Cas9 (*n* = 94)		Degree Centrality	In-Degree Centrality	Out-Degree Centrality	Eigenvector Centrality	Betweenness Centrality	Closeness Centrality	VoteRank Centrality
CNV–CNV networks	CNV–CNV network (GSE28582, *n =* 100)	pos (61/94)	pos (61/94)	pos (61/94)	pos (9/94)	pos (94/94)	pos (39/94)	neg (93/94)
	CNV–CNV network (GSE31800, *n =* 271)	pos (94/94)	pos (94/94)	pos (94/94)	pos (94/94)	pos (94/94)	pos (94/94)	neg (94/94)
CNV-mediated GE networks	CNV-mediated GE network (GSE28582, *n =* 100)	neg (40/94)	neg (94/94)	pos (72/94)	neg (94/94)	-	neg (91/94)	neg (43/94)
	CNV-mediated GE network (GSE31800, *n =* 49)	pos (90/94)	neg (72/94)	pos (94/94)	neg (76/94)	pos (15/94)	neg (92/94)	neg (21/94)
mRNA co-expression networks	mRNA co-expression network (GSE28582, *n =* 100)	neg (94/94)	neg (94/94)	neg (94/94)	neg (94/94)	neg (12/94)	neg (94/94)	pos (94/94)
	mRNA co-expression network (GSE31800, *n =* 49)	-	-	-	neg (94/94)	pos (81/94)	neg (94/94)	pos (4/94)
mRNA co-expression networks in Xu’s LUAD tumors and NATs	mRNA co-expression network in LUAD tumors (*n =* 51)	neg (64/94)	neg (64/94)	neg (64/94)	neg (91/94)	pos (6/94)	neg (76/94)	-
	mRNA co-expression network in LUAD NATs (*n =* 49)	neg (94/94)	neg (94/94)	neg (94/94)	neg (94/94)	neg (38/94)	neg (94/94)	pos (92/94)
mRNA-mediated protein expression networks in Xu’s LUAD tumors and NATs	mRNA-mediated protein expression network in LUAD tumors (*n =* 51)	neg (94/94)	neg (94/94)	pos (94/94)	neg (94/94)	neg (94/94)	neg (94/94)	neg (94/94)
	mRNA-mediated protein expression network in LUAD NATs (*n =* 49)	neg (94/94)	neg (94/94)	pos (94/94)	neg (94/94)	neg (94/94)	neg (94/94)	neg (30/94)
Protein co-expression networks in Xu’s LUAD tumors and NATs	Protein co-expression network in LUAD tumors (*n =* 103)	-	-	-	-	-	-	neg (21/94)
	Protein co-expression network in LUAD NATs (*n =* 103)	neg (93/94)	neg (93/94)	neg (93/94)	neg (94/94)	-	neg (75/94)	-

**Table 5 biomolecules-12-01782-t005:** Correlations of seven centrality metrics of selected networks with RNAi dependency scores. pos: Pearson’s correlation coefficient *r* > 0 and *p* < 0.05; neg: *r* < 0 and *p* < 0.05. The numbers in parentheses showed the number of significant NSCLC cell lines. The blue color indicates concordant positive correlations. The orange color indicates concordant negative correlations.

Proliferation—RNAi (*n* = 92)		Degree Centrality	In-Degree Centrality	Out-Degree Centrality	Eigenvector Centrality	Betweenness Centrality	Closeness Centrality	VoteRank Centrality
CNV–CNV networks	CNV–CNV network (GSE28582, *n =* 100)	pos (17/92)	pos (17/92)	pos (17/92)	-	pos (88/92)	pos (8/92)	neg (91/92)
	CNV–CNV network (GSE31800, *n =* 271)	pos (66/92)	pos (66/92)	pos (66/92)	pos (70/92)	pos (82/92)	pos (67/92)	neg (5/92)
CNV-mediated GE networks	CNV-mediated GE network (GSE28582, *n =* 100)	neg (27/92)	neg (92/92)	pos (11/92)	neg (92/92)	-	neg (91/92)	-
	CNV-mediated GE network (GSE31800, *n =* 49)	pos (1/92)	neg (12/92)	pos (9/92)	neg (9/92)	pos (1/92)	neg (65/92)	pos (1/92)
mRNA co-expression networks	mRNA co-expression network (GSE28582, *n =* 100)	neg (92/92)	neg (92/92)	neg (92/92)	neg (92/92)	neg (80/92)	neg (92/92)	pos (92/92)
	mRNA co-expression network (GSE31800, *n =* 49)	neg (24/92)	neg (24/92)	neg (24/92)	neg (74/92)	pos (2/92)	neg (92/92)	pos (6/92)
mRNA co-expression networks in Xu’s LUAD tumors and NATs	mRNA co-expression network in LUAD tumors (*n =* 51)	-	-	-	neg (1/92)	pos (33/92)	-	neg (27/92)
	mRNA co-expression network in LUAD NATs (*n =* 49)	neg (92/92)	neg (92/92)	neg (92/92)	neg (92/92)	neg (82/92)	neg (92/92)	pos (78/92)
mRNA-mediated protein expression networks in Xu’s LUAD tumors and NATs	mRNA-mediated protein expression network in LUAD tumors (*n =* 51)	neg (92/92)	neg (92/92)	pos (92/92)	neg (92/92)	neg (92/92)	neg (92/92)	neg (89/92)
	mRNA-mediated protein expression network in LUAD NATs (*n =* 49)	neg (92/92)	neg (92/92)	pos (92/92)	neg (92/92)	neg (88/92)	neg (92/92)	neg (1/92)
Protein co-expression networks in Xu’s LUAD tumors and NATs	Protein co-expression network in LUAD tumors (*n =* 103)	-	-	-	neg (1/92)	-	-	neg (10/92)
	Protein co-expression network in LUAD NATs (*n =* 103)	neg (32/92)	neg (32/92)	neg (32/92)	neg (65/92)	neg (1/92)	neg (14/92)	-

**Table 6 biomolecules-12-01782-t006:** Correlations of seven centrality metrics of selected networks with hazard ratios in univariate Cox modeling of combined TCGA–LUAD (*n* = 515) and TCGA–LUSC (*n* = 501). pos: Pearson’s correlation coefficient *r* > 0 and *p* < 0.05; neg: *r* < 0 and *p* < 0.05. The blue color indicates concordant positive correlations. The orange color indicates concordant negative correlations.

Patient Survival—Hazard Ratio in Combined TCGA–LUAD (*n* = 515) and TCGA–LUSC (*n* = 501)		Degree Centrality	In-degree Centrality	Out-Degree Centrality	Eigenvector Centrality	Betweenness Centrality	Closeness Centrality	VoteRank Centrality
CNV–CNV networks	CNV–CNV network (GSE28582, *n =* 100)	-	-	-	-	-	-	-
	CNV–CNV network (GSE31800, *n =* 271)	neg	neg	neg	neg	neg	neg	pos
CNV-mediated GE networks	CNV-mediated GE network (GSE28582, *n =* 100)	-	pos	-	pos	-	-	-
	CNV-mediated GE network (GSE31800, *n =* 49)	neg	-	neg	-	-	-	-
mRNA co-expression networks	mRNA co-expression network (GSE28582, *n =* 100)	pos	pos	pos	pos	-	pos	neg
	mRNA co-expression network (GSE31800, *n =* 49)	pos	pos	pos	pos	-	pos	-
mRNA co-expression networks in Xu’s LUAD tumors and NATs	mRNA co-expression network in LUAD tumors (*n =* 51)	neg	neg	neg	neg	neg	neg	pos
	mRNA co-expression network in LUAD NATs (*n =* 49)	-	-	-	-	-	-	pos
mRNA-mediated protein expression networks in Xu’s LUAD tumors and NATs	mRNA-mediated protein expression network in LUAD tumors (*n =* 51)	pos	pos	neg	pos	pos	pos	-
	mRNA-mediated protein expression network in LUAD NATs (*n =* 49)	pos	pos	neg	pos	pos	pos	-
Protein co-expression networks in Xu’s LUAD tumors and NATs	Protein co-expression network in LUAD tumors (*n =* 103)	-	-	-	pos	-	-	-
	Protein co-expression network in LUAD NATs (*n =* 103)	pos	pos	pos	pos	pos	pos	neg

**Table 7 biomolecules-12-01782-t007:** Selected hub genes that were significant in all measurements of tumorigenesis and patient survival. LUAD: Xu’s LUAD cohort [39]. DE: differential expression. Fold change: tumor/NATs. The percentage in proliferation results represented the number of cell lines with a dependency score < −0.5 divided by the total number of tested human NSCLC cell lines in CRISPR-Cas9/RNAi screening. CI: confidence interval.

Gene Name	mRNA DE *t* Statistics in LUAD	mRNA DE Fold Change in LUAD	Protein DE *t* Statistics in LUAD	Protein DE Fold Change in LUAD	Proliferation% (CRISPR-Cas9)	Proliferation% (RNAi)	Survival Hazard Ratio in TCGA	95% CI of Survival Hazard Ratio in TCGA
*BUB3*	10.45	1.78	12.50	1.05	94/94	2/92	1.26	[1.02, 1.55]
*DNM1L*	3.10	1.29	14.21	1.08	50/94	41/92	1.17	[1.01, 1.37]
*EIF2S1*	5.84	1.60	15.79	1.04	94/94	70/92	1.24	[1.03, 1.5]
*GALNT2*	6.50	1.92	15.87	1.10	0/94	0/92	1.28	[1.12, 1.47]
*KPNB1*	9.56	1.80	14.58	1.04	94/94	73/92	1.24	[1.01, 1.52]
*NMT1*	8.36	1.46	17.63	1.10	62/94	0/92	1.31	[1, 1.71]
*PFKP*	6.09	2.51	16.59	1.10	1/94	0/92	1.21	[1.09, 1.34]
*PGAM1*	3.81	1.48	20.62	1.09	94/94	1/92	1.17	[1.01, 1.35]
*PTGES3*	5.79	1.49	17.46	1.10	3/94	0/92	1.24	[1.03, 1.49]
*STRAP*	5.61	1.66	18.41	1.09	84/94	8/92	1.17	[1.02, 1.35]

## Data Availability

Data access is provided in the manuscript. An implementation of our Boolean implication networks is available in SouceForge: https://sourceforge.net/p/genet-cnv/activity/?page=0&limit=100#6158cc8bbdf93eb6c5d200d8. Since we patented our AI technology and software for discovery of biomarkers and therapeutic targets (PCT/US22/75136), the current version of the software is not released.

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
