# Peer review of "Hub Genes in Non-Small Cell Lung Cancer Regulatory Networks"

_biomolecules, 2022, doi:10.3390/biom12121782_

Round 1
Reviewer 1 Report
In this manuscript, the authors employed system biology approaches to identify hub genes in NSCLC. By combining several data sets, the authors constructed multi-omics network. The methods applied are valid. Thus, this reviewer has only few comments to make:
Major points:
[1] Both GSE28582 and GSE31800 are microarray data of different designs. First, how did the authors handle the missing genes (present in one microarray platform but not the other)? Second, no information is provided as to how the microarray data are processed, including pre-processing and normalization.
[2] Lines 246 - 250: "To test if a constructed Boolean implication network had higher average centrality values compared with random networks, the averaged centrality metrics (except VoteRank centrality) of the constructed network were compared with those of 1,000 randomly selected networks with the same number of genes."
Lines 411 - 413: "Here, we examined if these two clinically relevant multi-omics networks had higher network centralities in the genome-scale compared with 1,000 random networks with the same number of genes."
No information is provided which method was used to select such "random" networks.
Minor points:
(1) Line 102: "The genome reference version was converted to hg38." Which version of hg38 did the authors use?
(2) It is now a common practice to disclose all the computer algorithms, commands, and pipelines via a public domain, such as GitHub.
(3) One- or two-tail t-test? Paired or unpaired?
Author Response
Comments and Suggestions for Authors
Reviewer: In this manuscript, the authors employed system biology approaches to identify hub genes in NSCLC. By combining several data sets, the authors constructed multi-omics network. The methods applied are valid. Thus, this reviewer has only few comments to make:
Authors: We thank the reviewer for all the constructive comments.
Major points:
Reviewer: [1] Both GSE28582 and GSE31800 are microarray data of different designs. First, how did the authors handle the missing genes (present in one microarray platform but not the other)? Second, no information is provided as to how the microarray data are processed, including pre-processing and normalization.
Authors: All the available genes in GSE28582 and GSE31800, respectively, were first used to construct genome-scale networks and then to compute the correlations between network centralities and tumorigenesis, proliferation, and patient survival. The focus of the analysis here is the association between network centralities and clinical phenotypes. Missing genes in each microarray platform were not included in the computation of the correlation coefficients in the genome scale. The results with significant Pearson’s correlation coefficients with the same sign in both datasets were considered concordant.
GSE31800: Gene expression data were generated using the Custom Rosetta-Affymetrix Human platform. Fresh-frozen lung tumors were obtained from Vancouver General Hospital. Microdissection of tumor cells was performed, and total RNA was isolated using RNeasy Mini Kits (Qiagen Inc.). Samples were labeled and hybridized to a custom Affymetrix microarray, containing 43,737 probes mapping to approximately 23,000 unique genes, according to the manufacturer's protocols (Affymetrix Inc.) All data were normalized using the Robust Multichip Average algorithm in R. Of the lung tumor cohort, only samples with sufficient material for RNA isolation were selected for expression analysis.
GSE28582: A total of 2 μg RNA (RIN value >7.0) from each tissue specimen was used for analysis on Affymetrix Human Genome U133 Plus 2 arrays (Affymetrix Inc.). Sample preparation, processing, and hybridization were performed according to the GeneChip Expression Analysis Technical Manual (Affymetrix Inc., Rev. 5). Subsequent analyses of the gene expression data were carried out in the freely available statistical computing language R (http://www.r-project.org) using packages available from the Bioconductor project. The raw data were normalized using the robust multiarray average method and have been uploaded together with clinical information on GEO with the accession number GSE28582 (www.ncbi.nlm.nih.gov/geo/). Only transcripts with average signal intensities above 5 were used for further analysis. For the comparison of gene expression levels between different patient groups, a two-sided Student's t-test was used.
The above information is now added to the manuscript Sections 2.2.1 and 2.2.2.
Reviewer: [2] Lines 246 - 250: "To test if a constructed Boolean implication network had higher average centrality values compared with random networks, the averaged centrality metrics (except VoteRank centrality) of the constructed network were compared with those of 1,000 randomly selected networks with the same number of genes."
Lines 411 - 413: "Here, we examined if these two clinically relevant multi-omics networks had higher network centralities in the genome-scale compared with 1,000 random networks with the same number of genes."
No information is provided which method was used to select such "random" networks.
Authors: The random networks contained the same number of genes randomly selected from the whole genome excluding our identified network genes. This is now added in Statistical Methods (Section 2.6).
Minor points:
Reviewer: (1) Line 102: "The genome reference version was converted to hg38." Which version of hg38 did the authors use?
Authors: The genome annotation was obtained from the UCSC genome browser with the Python package cruzdb on January 28, 2020. This information is now added in Section 2.2.1.
Reviewer: (2) It is now a common practice to disclose all the computer algorithms, commands, and pipelines via a public domain, such as GitHub.
Authors: An implementation of our Boolean implication networks is available in SouceForge: https://sourceforge.net/p/genet-cnv/activity/?page=0&limit=100#6158cc8bbdf93eb6c5d200d8
Since we patented our AI technology and software for the discovery of biomarkers and therapeutic targets (PCT/US22/75136), the current version of the software is not released. This information is now provided in Data Availability Statement in the manuscript.
Reviewer: (3) One- or two-tail t-test? Paired or unpaired?
Authors: We used two-tailed two-sample t-tests (unpaired). It is now clarified in Statistical Methods (Section 2.6).
Reviewer 2 Report
The topic is research is interesting and the paper is well organized. However, there are a few points has to be corrected.
1) Although the manuscript is a computational work, the abstract is lack numerical results.
2) You mentioned some important genes in NSCLC including, BUB3 DNM1L, EIF2S1, KPNB1, NMT1, PGAM1, and PGAM1, however, these genes do not work alone. It would be more interesting if the authors make a veen diagram to show the relationship between them to make NSCLC.
3) CD151 is one of the biomarkers in NSCLC, why it has not been studied in fig2?
4) Although the studying of Hub genes is interesting, how it can be used in clinical diagnosis?
Author Response
Reviewer 2
Comments and Suggestions for Authors
Reviewer: The topic is research is interesting and the paper is well organized. However, there are a few points has to be corrected.
Authors: We thank the reviewer for all the constructive comments.
Reviewer: 1) Although the manuscript is a computational work, the abstract is lack numerical results.
Authors: Statistical results are now added in the Abstract.
Reviewer: 2) You mentioned some important genes in NSCLC including, BUB3 DNM1L, EIF2S1, KPNB1, NMT1, PGAM1, and PGAM1, however, these genes do not work alone. It would be more interesting if the authors make a veen diagram to show the relationship between them to make NSCLC.
Authors: Venn diagrams of relationships of gene associations involving these genes in NSCLC are now provided in Supplementary file 5. A network of genes and proteins that had a significant association (p < 0.05, z tests) with all these genes in NSCLC patient tumors is now provided in Figure 3 in the manuscript. The enriched gene families and cytobands of this network are included in Supplementary file 6.
Reviewer: 3) CD151 is one of the biomarkers in NSCLC, why it has not been studied in fig2?
Authors: To show the relevance of network centrality and therapeutic targets, Figure 2 included immunotherapy targets that are either used for treating NSCLC patients (including PD1, PDL1, and CTLA4) or showed promising results in phase I/II clinical trials (CD27). Many NSCLC biomarkers including CD151 that have not been substantiated in clinical trials were not included in Figure 2. CD151 was ranked as a top hub gene within the 10th percentile of degree centrality in the CNV-mediated gene expression network in GSE28582 and
mRNA-mediated protein expression network in Xu's LUAD NATs. This is now added to the Discussion.
Reviewer: 4) Although the studying of Hub genes is interesting, how it can be used in clinical diagnosis?
Authors: This study shows that multi-omics network centrality can be used as a prioritization method in the selection of biomarkers and therapeutic targets. Hub genes can be candidate genes for the development of clinical diagnostic tests. The final determination of the inclusion of the candidate genes in the clinical tests will be made based on the assay optimization and validation results in multiple patient cohorts according to REMARK guidelines [1, 2]. This is now added to the Discussion.
References
- Jankova, L., et al., Reporting in studies of protein biomarkers of prognosis in colorectal cancer in relation to the REMARK guidelines. Proteomics Clin Appl, 2015. 9(11-12): p. 1078-86.
- McShane, L.M., et al., Reporting recommendations for tumor marker prognostic studies (REMARK). J. Natl. Cancer Inst, 2005. 97(16): p. 1180-1184.
Round 2
Reviewer 2 Report
The authors provided the answers to my comments and I recommend publishing.